# Menstrual flow as a non-invasive source of endometrial organoids

Tereza Cindrova-Davies [1✉], Xiaohui Zhao [1], Kay Elder[2], Carolyn J. P. Jones [3], Ashley Moffett [1,4], Graham J. Burton [1] & Margherita Y. Turco [1,4]

Assessment of the endometrium often necessitates a biopsy, which currently involves an invasive, transcervical procedure. Here, we present an alternative technique based on deriving organoids from menstrual flow. We demonstrate that organoids can be derived from gland fragments recovered from menstrual flow. To confirm they faithfully reflect the in vivo state we compared organoids derived from paired scratch biopsies and ensuing menstrual flow from patients undergoing in vitro fertilisation (IVF). We demonstrate that the two sets of organoids share the same transcriptome signature, derivation efficiency and proliferation rate. Furthermore, they respond similarly to sex steroids and early-pregnancy hormones, with changes in morphology, receptor expression, and production of 'uterine milk' proteins that mimic those during the late-secretory phase and early pregnancy. This technique has wide-ranging impact for non-invasive investigation and personalised approaches to treatment of common gynaecological conditions, such as endometriosis, and reproductive disorders, including failed implantation after IVF and recurrent miscarriage.

[1] Centre for Trophoblast Research, Department of Physiology, Development and Neuroscience, University of Cambridge, Cambridge, UK. [2] Bourn Hall Clinic, Cambridge, UK. [3] Maternal and Fetal Health Research Centre Division of Developmental Biology & Medicine School of Medical Sciences Faculty of Biology, Medicine and Health University of Manchester, Manchester Academic Health Science Centre, Manchester, UK. [4] Department of Pathology, University of Cambridge, Cambridge, UK. ✉email: tc269@cam.ac.uk

Organoids are self-organising, three-dimensional culture systems that can be cultured long-term and faithfully reflect the structure and function of their tissue of origin. They have now been derived from many tissues, and have proved powerful tools for the investigation of normal development and modelling disease[1]. Their physiological responsiveness enables them to be used to predict drug effects, and they open new avenues for personalised regenerative medicine.

We, and others, have previously shown that endometrial organoids provide a valuable model for investigating maternal-fetal interactions during early pregnancy, and for exploring the pathophysiology of gynaecological complications such as endometriosis and endometrial cancer[2–5]. To date, the organoids have been derived from biopsy samples, which, while reproducible and robust, necessitate an invasive procedure performed by a trained clinician. We therefore determined whether organoids can be derived non-invasively from menstrual flow. To assess whether these faithfully replicate the molecular signature of the endometrium in vivo, we adopted a unique experimental approach whereby women undergoing an endometrial scratch as part of an in vitro fertilisation (IVF) procedure were invited to collect menstrual flow from the same cycle using an endometrial cup.

## Results

**Optimisation of the protocol.** We performed an initial pilot study to optimise the technique for derivation of organoids from menstrual flow. We received samples of flow collected in a menstrual cup from seven volunteers with normal cycles, and succeeded in deriving organoids from six. A repeat sample from the volunteer whose culture was unsuccessful also failed. Cells isolated from the flow appeared dead, and we attributed this to sample collection/volunteer problems and did not explore the issue further. A further three volunteers provided repeat samples, and we successfully derived organoids from all. The detailed protocol is given in the Methods, and was followed for the derivation of organoids from clinical samples.

**Growth characteristics of organoids derived from menstrual flow.** In order to validate the menstrual flow organoids, we derived organoids from endometrial scratches and ensuing menstrual flow of volunteers undergoing an IVF treatment cycle at the Bourn Hall Clinic (Fig. 1a). We received menstrual samples from eight patients and succeeded in deriving organoids from seven. In the failed sample, the starting tissue appeared white and dead, but the cause was not explored further. Altogether, the success rate of deriving organoids from menstrual flow was 87%.

The rate of growth varied between patients, as it does in organoids derived from pipelle biopsies[3] (Fig. 1b–d and Supplementary Fig. 1). We were also able to freeze menstrual gland digest samples and recover them later (Fig. 1b and Supplementary Fig. 1). Organoids obtained from menstrual blood vs. scratch biopsies were indistinguishable morphologically (Fig. 1e). In most cases, the initial yield of tissue fragments from menstrual flow was lower than that obtained from scratch biopsies. However, the speed of organoid establishment was comparable to that from endometrial pipelle biopsies[3]. To compare the growth rate of menstrual vs. scratch organoids, we quantified organoid efficiency formation in paired single-cell

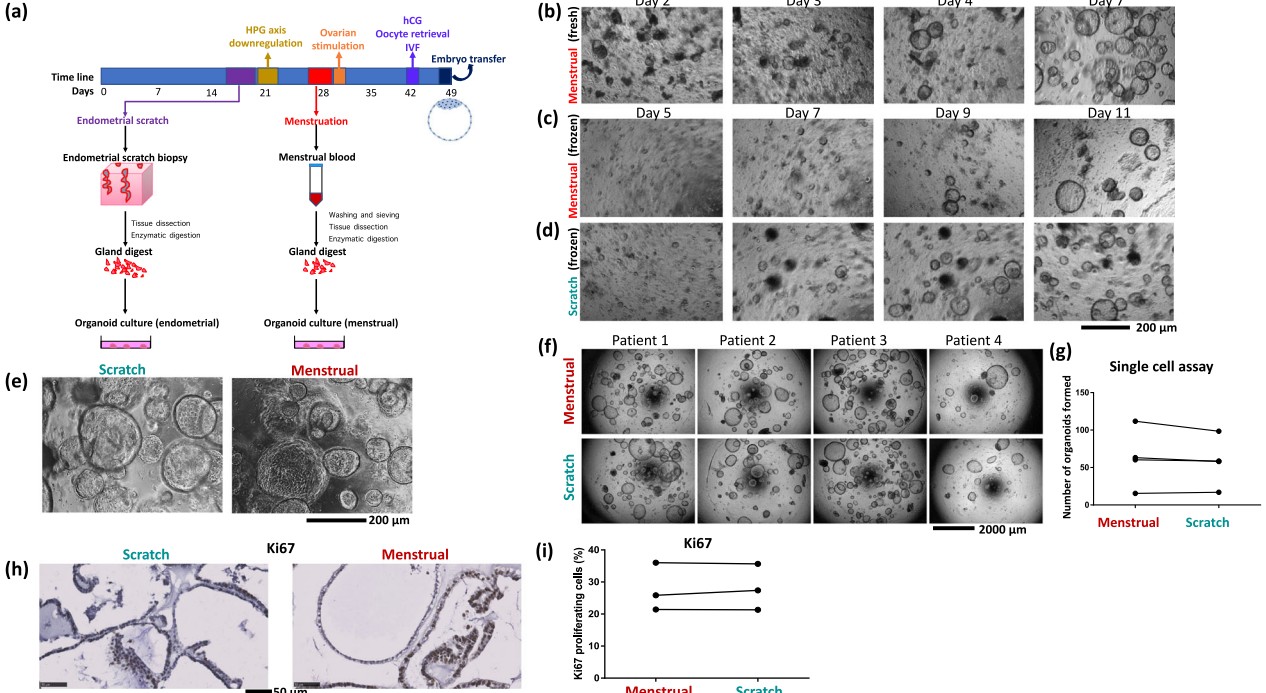

**Fig. 1 Derivation and characterisation of menstrual flow organoids. a** Schematic to illustrate how organoids were derived from an endometrial scratch and menstrual flow in the same cycle. **b–d** Time line images detailing propagation of organoids from menstrual flow and scratch. **b** The panel represents the growth of menstrual organoids directly derived from menstrual blood. **c, d** The panels illustrate the growth of organoids from frozen menstrual (**c**) and scratch (**d**) digests of the same patient. Scale bar 400 μm. **e** Organoids derived from an endometrial scratch and menstrual flow of the same patient are indistinguishable morphologically. Scale bar 200 μm. **f, g** Single-cell assay to compare the growth of paired menstrual vs. scratch organoids derived from four patients. Images show organoids seeded from single cells (5000 cells per 20 μl drop Matrigel) after 9 days of growth (**f**). Quantification of the single-cell assay. Scale bar 2000 μm. Five to 8-well repeats were quantified per condition (**g**). **h** Representative images of immunostaining against the marker of proliferation, Ki67, in menstrual and scratch organoids of one patient. Scale bar 50 μm. **i** Quantification of the Ki67 proliferation index in menstrual vs. scratch organoids of three patients.

assays and observed no difference (mean ± SE organoid efficiency formation: 62.67 ± 19.70 menstrual organoids vs. 57.97 ± 16.64 scratch organoids formed after 9 days from 5000 single cells seeded per well, $p = 0.86$) (Fig. 1f, g). We also assessed organoid proliferation rate using Ki67 staining (Fig. 1h, i), and again found no difference (mean ± SE Ki67 proliferation rate: 27.76 ± 4.32% menstrual vs. 28.12 ± 4.16% scratch; $p = 0.95$). Menstrual flow organoids began to form after 2–4 days in culture if grown from fresh samples (Fig. 1b), or after 5 days if grown from frozen digests (Fig. 1c, d and Supplementary Fig. 1), and were ready to passage after 7–10 days (Fig. 1b, d and Supplementary Fig. 1).

**Characterisation of organoids derived from menstrual flow.** RNA sequencing (RNA-Seq) revealed that organoids derived from scratch biopsies and menstrual flows clustered together (Fig. 2a). In addition, each patient's organoids paired up, showing conservation of transcripts across the two sources (Fig. 2b). The menstrual flow organoids expressed a transcript signature rich in markers of progenitor cells (*LRIG1, PROM1, AXIN2, SOX9*), proliferation (*MKI67, PCNA, TOP2A*), epithelial cell lineage (*EPCAM, KRT7, CLDN10, CDH1*), endometrial gland development and function (*FOXA2, KLF5, SOX17*), epithelial secretory activity (*PAEP, MUC1, MUC20, PAX8, KLK11, SPP1*), receptors to pregnancy hormones (*PGR, PRLR, PRLHR, GHR, ESRRA, ESRRB, ESRRG, ESR2*), and cilia formation (*FOXJ1, PIFO, RSPH1*), in an identical pattern to organoids derived from the scratch biopsies (Fig. 2c).

To validate the physiological relevance of the menstrual flow organoids, we tested their responsiveness to treatment with early-pregnancy hormones and found they reacted in an identical fashion to those derived from scratch biopsies from the same patient. Treatment with progesterone, oestrogen, cAMP, as well as prolactin (PRL), human placental lactogen (hPL) and human chorionic gonadotropin (hCG) increased the mRNAs encoding PAEP (glycodelin, GdA), MUC1, and LIF (Fig. 2d), as well as PAEP/glycodelin, and MUC1 protein expression (Fig. 2e, f), and increased secretion of glycodelin in culture supernatants (Fig. 2g) of both menstrual and scratch organoids.

**Response of organoids to sex steroids and early-pregnancy hormones.** Administration of progesterone, oestrogen, cAMP, PRL, hPL and hCG also increased expression of acetylated tubulin, P4-R, and PRL-R (Fig. 3a). We explored the changes in P4-R further. Combined hormonal treatment induced an increase in P4-R expression in menstrual and scratch organoids when compared to the control medium culture, in agreement with previous reports[3] (Fig. 3a–c). However, treatment with E2 alone induced an extensive increase in P4-R expression, which was suppressed when P4 was added to the culture medium of both menstrual and scratch organoids (Fig. 3d). These results are in line with reports of increased progesterone receptor during the proliferative phase of the cycle (E2 dominant), and its down-regulation during the secretory phase and early pregnancy (P4 dominant)[6].

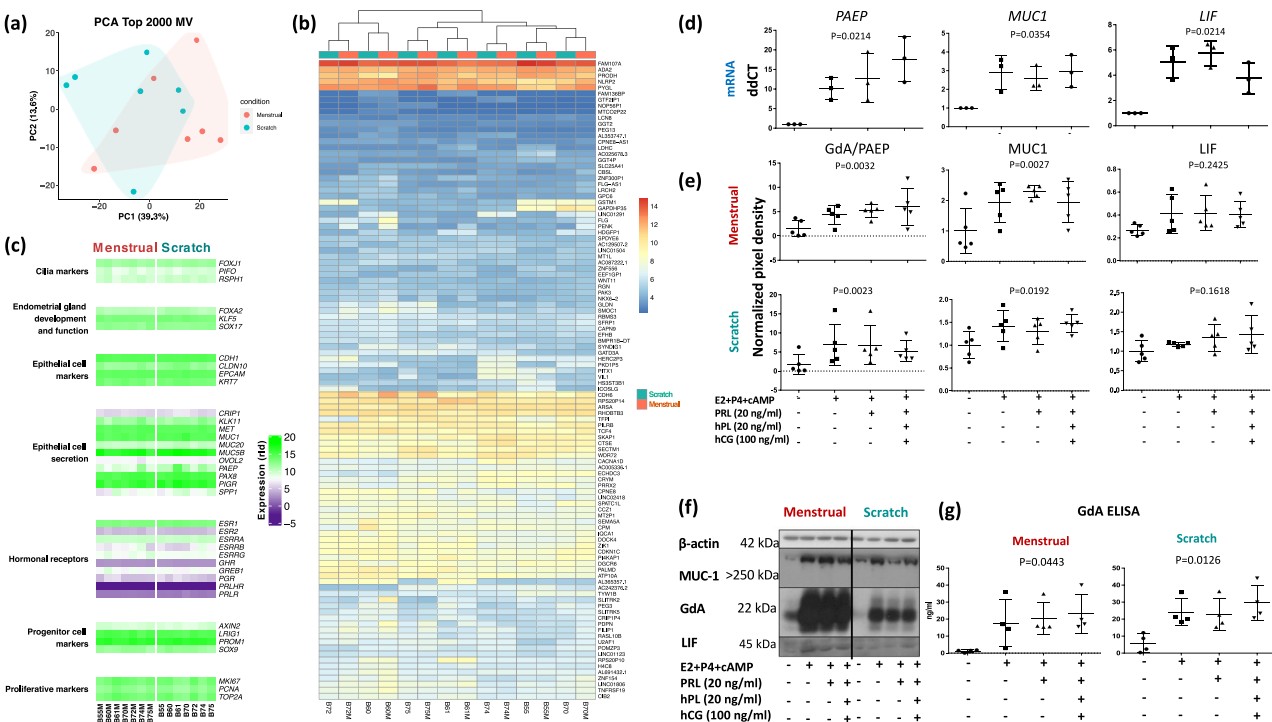

**Fig. 2 Comparative characterisation of menstrual and scratch organoids and their response to hormones. a** PCA plot shows top 2000 variable genes clustering of organoids derived from scratch biopsies and menstrual flow of seven patients analysed by RNA-Seq. **b** A heatmap of scratch and menstrual organoids demonstrates pairing of organoids in each of seven patients using the 101 differential expressed genes with threshold $p_{adj} < 0.05$, abs(log2Fold-Change) >= 1. **c** A heatmap validating expression of epithelial, secretory and stem cell markers in scratch and menstrual organoids. **d, g** Menstrual flow organoids are hormonally responsive. Organoids were grown for 4 days, and treated with culture medium alone or β-estradiol for 2 days followed by β-estradiol, progesterone and cAMP (EPC), EPC plus prolactin, or EPC plus prolactin, hPL and hCG for 4 days. **d** mRNA of *PAEP, LIF* and *MUC1* was quantified using qRT-PCR in hormonally treated menstrual organoids ($n = 3$). Statistical significance was assessed using ANOVA, $p$-values are shown on each graph. **e, f** Menstrual flow and scratch organoids ($n = 5$ each) responded similarly to hormones at the protein level. PAEP/glycodelin, MUC1 and LIF were detected by western blotting. **g** Quantification of secreted glycodelin detected in the supernatants of menstrual and scratch organoids ($n = 4$ each) treated with hormones. Statistical significance was assessed using ANOVA, $p$-values are shown on each graph, error bars represent standard deviation.

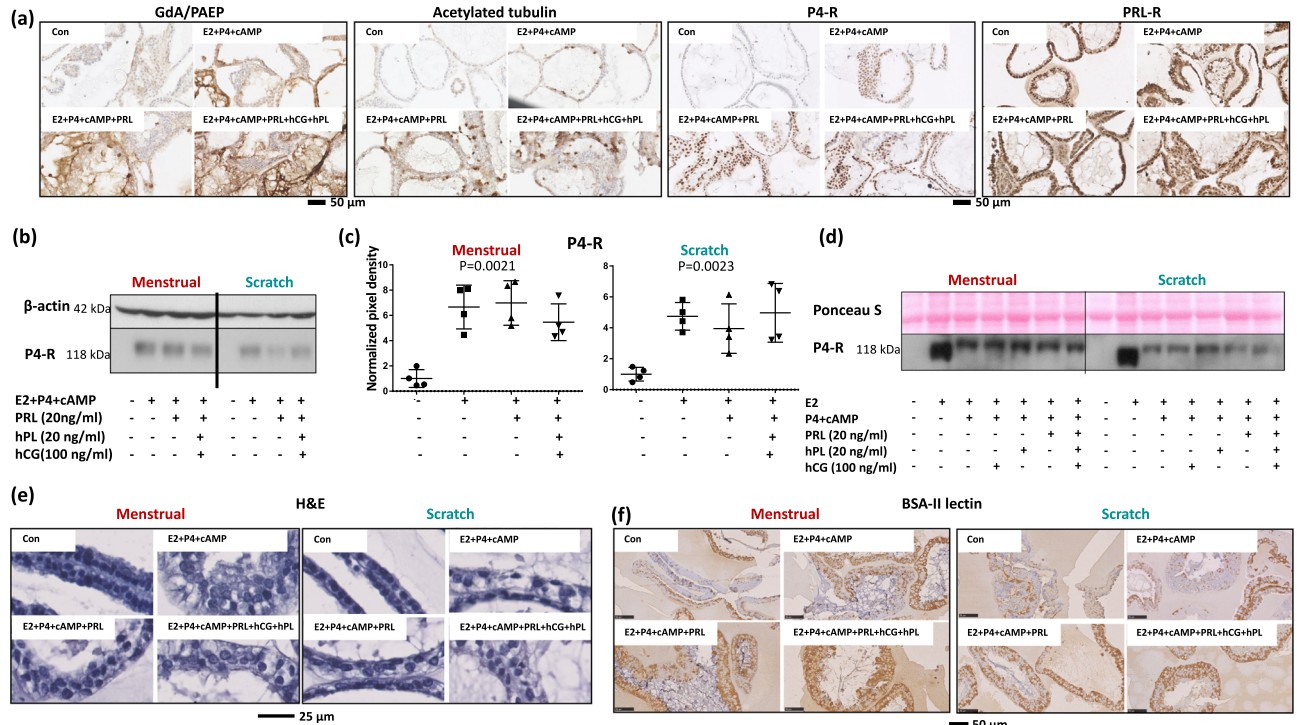

**Fig. 3 Morphological changes in menstrual and scratch organoids in response to hormones. a** Menstrual organoids show immunohistochemical upregulation of PAEP/GdA, acetylated tubulin, progesterone receptor (P4-R) and prolactin receptor (PRL-R) in response to hormones as organoids derived from pipelle biopsies. Scale bar 50 μm. **b**, **c** Treatment of menstrual and scratch organoids ($n = 4$ each) with different pregnancy hormones increased the expression of the progesterone receptor (P4-R), as detected by western blotting (**a**) and quantified (**b**). Statistical significance was assessed using ANOVA, $p$-values are shown on each graph, error bars represent standard deviation. **d** Representative western blots of menstrual and scratch organoids treated with a range of pregnancy hormones, including oestrogen alone (E2), which induced major upregulation of the P4-R expression. Adding of P4 alone or in combination with other hormones reduced E2-induced P4-R expression. **e** Representative images of haematoxylin and eosin (H&E) staining of hormonally treated menstrual and scratch organoids from one patient, showing morphological differences between untreated vs. hormonally treated organoids. Scale bar 25 μm. **f** *Bandeiraea simplicifolia* -II (BSA-II) lectin staining of semi-thin resin sections of menstrual and scratch organoids treated with hormones. Scale bar 50 μm.

In addition to inducing the formation of ciliated cells, as marked by increased expression of acetylated tubulin (Fig. 3a), application of steroid hormones alone, or in combination with PRL, hPL and hCG, induced a columnar epithelial morphology with increased vacuole formation analogous to the hypersecretory phenotype of early pregnancy (Fig. 3e, f and Supplementary Figs. 2 and 3)[7]. Increased staining of semi-thin resin sections with the lectins *Bandeiraea simplicifolia* -II (BSA-II) (Fig. 3f) and *Arachis hypogaea* agglutinin (AHA) was observed in both sets of organoids following treatment with hormones, in agreement with upregulation of glycoproteins in gland epithelial cells during the mid-secretory phase of the cycle and early pregnancy[8,9] (Fig. 3f and Supplementary Fig. 3). In addition, there was evidence of increased glycogen secretion by hormonally challenged menstrual and scratch organoids, as detected by the BSA-II lectin (Fig. 3f, BSA-II binds to amylase-sensitive glycogen) and by PAS staining (Supplementary Fig. 4).

## Discussion

Our results demonstrate that menstrual flow contains viable endometrial gland progenitor cells, and can be used to derive physiologically relevant organoid cultures in a non-invasive manner. We confirmed that the transcriptome of the resultant organoids is identical to that of organoids generated from an endometrial biopsy taken earlier in the same cycle. Furthermore, the two sets of organoids respond in the same fashion to steroid and early-pregnancy hormones, upregulating expression and

secretion of 'uterine milk' proteins. Together, our findings validate the use of menstrual flow as an easily obtainable and non-invasive alternative to endometrial biopsies for the derivation of endometrial organoids.

The human endometrium is a uniquely dynamic tissue controlled by the hypothalamic-pituitary-gonadal axis, mediated by a sequential interplay of sex hormones during the menstrual cycle. Following menstruation, the oestrogen-dominated proliferative phase promotes regrowth of the mucosa, which then differentiates and increases in functional activity during the progesterone-dominated secretory phase. Endometrial glands open onto the surface epithelium of the uterus that comprises a mix of ciliated and non-ciliated cuboidal cells. The glands are of the simple tubular form, interconnected basally[10], and are lined by a columnar epithelium. In preparation for implantation, the glands and intervening stromal cells undergo differentiation and an increase in secretory activity. The hormonal microenvironment coupled with secretions from the glands regulates the receptivity of the endometrial epithelium and the implantation competency of the blastocyst[11,12]. If pregnancy does not ensue, shedding occurs at menstruation, with regeneration during the proliferative phase of the next cycle. Our results demonstrate that both the menstrual and scratch organoids undergo equivalent changes in response to treatment with sex steroids.

A further increase in activity is seen during early pregnancy when the glands adopt a hypersecretory phenotype, the Arias-Stella reaction[7], thought to be induced by hormones released from the conceptus and decidua. In domestic species it is well

established that the conceptus stimulates and regulates its early development through a dialogue with the endometrial glands[13–15]. In particular, increased stimulation by progesterone, prolactin, and trophoblast-produced hormones leads to increased secretion of 'uterine milk' proteins and growth factors (e.g., EGF, LIF), which, in turn, stimulate trophoblast growth and differentiation. These effects also involve changes in receptor expression on the gland cells[13–15]. We speculate that a similar mechanism operates in the human[3,16] and the data presented here further support the case.

The 'uterine milk' proteins provide essential histotrophic nutrition for the conceptus during the first trimester of pregnancy before the definitive haemochorial placenta is established[17]. The importance of histotroph during early pregnancy has been demonstrated in animal models. Suppression of gland development in the sheep and mouse leads to failure of the conceptus to thrive and implant[18,19] As a corollary in the human, deficient gland function may contribute to poor placental development and early-pregnancy loss. For example, the composition of uterine fluid during the secretory phase of the cycle is abnormal in women with idiopathic infertility compared to fertile controls[20]. Also, pilot studies indicate that levels of prolactin and the glycodelin A, one of the major components of histotroph, are abnormally low in women suffering implantation failure or early-pregnancy loss[21,22]. In the human, the prolactin that appears to be a powerful stimulant of gland activity during early pregnancy is secreted by the decidua rather than the trophoblast as in other species[23]. This may explain the mounting evidence linking defective decidualisation to miscarriage and other complications of pregnancy[24].

Investigating the physiology and function of the normal cycling endometrium is challenging due to its great variability between individuals[25], reflecting the profound effects of diverse factors, including BMI, age and parity. Exploring the impact of these factors on implantation and early pregnancy is even more difficult due to obvious ethical concerns and the lack of a suitable animal model. Generating organoids non-invasively from a broad population of women will provide essential data to fill these important knowledge gaps. The organoids can be frozen, biobanked and thawed, allowing for future large-scale retrospective studies. They can also be seeded on to three-dimensional collagen scaffolds, paving the way for the development of co-culture techniques with which to explore implantation[26]. Currently, endometrial organoids are being used to investigate physiological changes during the normal cycle[5], and pathological endometrial

dysfunction in endometriosis, endometrial cancer and other gynaecological conditions[2]. Implantation of the transferred embryo is a rate-limiting step in Assisted Reproductive Techniques (ART)[27], and organoids are enabling exploration of the molecular mechanisms underpinning implantation[4]. Our technique will enable patients undergoing fertility treatments to have their endometrial function assessed non-invasively prior to initiating a treatment cycle, allowing the hormonal regime and timing of treatment to be tailored accordingly. This approach provides opportunities for personalised medicine in a similar fashion to that of kidney organoids derived from the urine of patients[28].

The derivation of endometrial organoids from menstrual fluid, therefore, opens new avenues for assessing endometrial function in health and disease, and hence we anticipate has widespread applications in clinical practice.

## Methods

### Subjects

*Pilot study.* We recruited seven healthy volunteers for the pilot study with informed written consent (ethical approval HBREC.2017.10). Volunteers had normal cycles, and included nulliparous and parous individuals. Samples were collected on an anonymised basis.

*Bourn Hall Clinic patients.* Seven patients who undertook an elective endometrial scratch (ES) procedures prior to initiation of a treatment cycle at Bourn Hall Clinic, Cambridge, agreed to collect a sample of the menstrual flow that subsequently followed the ES procedure. Endometrial gland organoids were derived from a tissue sample taken during the ES procedure, as well as from a sample of menstrual flow. All procedures were carried out with informed written consent, and local ethics approval (East of England—Cambridge Central Research Ethics Committee 17/EE/0151). Patients' clinical details are listed in Table 1.

### Hormonal stimulation and IVF procedures. 

The endometrial scratch procedure (ES) was performed during the luteal phase of the cycle immediately prior to the treatment cycle. Six patients initiated scheduled treatments by inducing a withdrawal bleed after the oral contraceptive pill; one patient started luteal phase downregulation with a GnRH agonist the day after the ES procedure.

Stimulation for the treatment cycle was initiated within a few days after menstruation; patients who were treated with an OCP/Antagonist regime started gonadotropin stimulation on day 2 or 3. The patient who was treated with a luteal phase downregulation protocol started stimulation 21 days following her menstrual bleed. Blastocysts were transferred on day 5 after OCR. Two patients responded poorly to stimulation, with only one (B75) or two (B55) oocytes fertilised; their cleavage stage embryos were transferred on Day 2 after OCR. Two further patients were treated with a hormone replacement regime for endometrial preparation prior to scheduled transfer of blastocysts vitrified in a previous cycle; blastocysts were warmed and transferred 5 days after presumed ovulation.

---

**Table 1 Clinical details.**

| Sample ID | Outcome | Previous obstetric history | BMI | Age | Treatment | Drug protocol |
|---|---|---|---|---|---|---|
| B55 | Not pregnant | Para 0 + 0 | 30.48 | 39 | ICSI; two cleavage stage embryos transferred (2c, 4c) on Day 2 | Antagonist + gonadotrophin |
| B60 | Elective CS @ 37 wks, Twins (2900 g & 2300 g) | Para 1 + 0 | 17.01 | 39 | ICSI; two blastocysts transferred on Day 5 | Agonist + gonadotrophin |
| B61 | Not pregnant | Para 0 + 0 | 26.4 | 33 | FET Single BT, Day 5 | HRT |
| B70 | Live Birth, 38 + 2 wks (3657 g) | Para 0 + 2EPL | 19.96 | 27 | FET; two blastocysts transferred Day 5 | HRT |
| B72 | Not pregnant | Para 1 + 0 | 25.47 | 37 | IVF; 2 compacting morulae transferred on D5 | Antagonist + gonadotrophin |
| B74 | Unknown | Para 1 + 1EPL | Not recorded | 49 | For IVF Rx in Cyprus | unknown |
| B75 | EPL @ 7 wks | Para 0 + 0 | 30.06 | 41 | ICSI; one 4-cell embryo transferred on Day 2 | Antagonist + gonadotrophin |

*EPL Early-pregnancy loss, FET frozen embryo transfer, BT blastocyst transfer.*

**Extraction of endometrial gland organoids from menstrual flow**. Patients were supplied with a reusable menstrual cup (Mooncup, type A or B depending on their age and/or obstetric history) and its proper use was explained. Patients were instructed to collect menstrual flow in a sterile 50-ml tube as soon as the flow became heavy on day 1–2. This was to be achieved by collecting the heavy flow sample during the day over at least 4 h (minimum ~ 7 ml blood), or overnight, and the sample was kept refrigerated. Menstrual samples were couriered to the lab and either processed immediately, or kept at 4 °C and processed the next day. Samples were spun down at $600 \times g$ for 5 min and washed with phosphate buffered saline (PBS) several times to remove red blood cells, discarding the bloody supernatant and collecting only the cellular material after every wash. The cellular sediment was subsequently passed through a 100-µm sieve (Corning, 431752) and washed with more PBS. The sieve was inverted over a Petri dish and retained cellular debris was backwashed from the sieve membranes, to collect the solid endometrial gland-containing material. Collected tissue debris was chopped with a scalpel into small 5 mm fragments and then digested in 20 ml 25 U/ml Dispase II (Sigma, D4693)/0.4 mg/ml collagenase V (Sigma, C-9263) solution in RPMI 1640 medium (Thermo Fisher Scientific, 21875-034)/10% FBS with gentle shaking at 37 °C for 20 min. The solution was then neutralised with 20 ml cold RPMI 1640/10% FBS medium and passed through a clean 100-µm sieve. The sieve was inverted again over a Petri dish and retained glandular elements were backwashed from sieve membranes, using vigorous bursts of medium from a disposable Pasteur pipette, pelleted by centrifugation and resuspended in ice-cold Matrigel (Corning, 536231) at a volume: volume ratio of 1:20. Twenty-five microlitre drops of Matrigel–cell suspension were plated into 48-well plates (Costar, 3548), allowed to set at 37 °C and overlaid with 250 µl organoid culture medium, as described previously[3]. See Supplementary Fig. 5 for more details. The medium was changed every 2–3 days and cultures were passaged by manual pipetting every 7–10 days.

**Derivation of endometrial organoids from endometrial scratch biopsies**. Endometrial gland organoids were derived from scratch biopsies, as previously described[3]. Briefly, endometrial scratch biopsies were dissected using scalpels into ~0.5 mm³ cubes and enzymatically digested in 20 ml solution of 1.25 U/ml dispase II (Sigma, D4693)/0.4 mg/ml collagenase V (Sigma, C-9263) in PRMI 1640 medium containing 10% FBS with gentle shaking at 37 °C for 20–30 min. The supernatant was neutralised with medium and passed through one or more 100 µm cell sieves (Corning, 431752) and the sieve washed several times with medium. The sieves were inverted over a Petri dish and retained glandular elements were backwashed from the sieve membranes, pelleted by centrifugation and resuspended in ice-cold Matrigel (Corning, 536231) at a ratio of 1:20 (vol:vol). Twenty-five microlitre drops of Matrigel–cell suspension were plated into 48-well plates (Costar, 3548), allowed to set at 37 °C and overlaid with 250 µl organoid Expansion Medium (ExM)[3]. The medium was changed every 2–3 days. Cultures were passaged by manual pipetting every 7–10 days.

**Freezing and recovery of menstrual and scratch organoids**. Menstrual and scratch organoids were frozen in the freezing medium containing 70% organoid culture medium, 20% FBS and 10% DMSO. All derived organoids, as well as the starting digests, were frozen and biobanked, starting from gland/menstrual digests (passage 0) to higher passages. The viability of the frozen cultures reflects that of the cultures before freezing. For all experiments presented in this paper, we thawed established organoid cultures (not frozen digests) of a low passage (1–3), expanded their numbers and used experimentally. The derivation of organoids from frozen menstrual and gland digests was only presented (Fig. 1c, d and Supplementary Fig. 1) to show this as an alternative to deriving organoids from fresh digests, and to demonstrate for one patient the ease and comparability of deriving organoids from menstrual blood vs. endometrial scratch.

**Organoid treatment and processing**. For the hormonal challenge, organoids were seeded and grown for 4 days, followed by treatment with culture medium alone, or with hormones, which consisted of pre-treatment with β-estradiol (10 nM) for 2 days, followed by treatment with β-estradiol, progesterone (1 µM) and cAMP (1 µM) (EPC), EPC plus prolactin (20 ng/ml), or EPC plus prolactin (20 ng/ml), hPL (20 ng/ml) and hCG (100 ng/ml) for 4 days. At the end of each experiment, organoids were removed from Matrigel using Cell Recovery solution and either stored at −80 °C for later RNA extraction, or processed for western blotting or immunohistochemistry, as detailed below.

**Organoid efficiency formation from single cells**. Menstrual and scratch organoids from four patients were grown in the organoid medium for 7 days. Experiments were performed between passage 2–6, and paired menstrual and scratch organoids from the same patient were derived at the same growth passage. After 7 days, organoids were removed from Matrigel using Cell Recovery Solution, pipetted several hundred times, trypsinized with Trypsin-EDTA for 4 min at 37 °C and washed in medium to neutralise the Trypsin. Cells were passed through a 40 µm cell strainer (Corning, 352340) to ensure single-cell suspension, and their numbers counted using a haemocytometer (trypan blue being used to exclude dead cells). Five-thousand cells were plated per 20 µl Matrigel drop into 48-well plate,

with 5–8-well repeats used per condition. The number of organoids formed after 9 days was scored.

**RNA sequencing**. RNA was extracted from organoids using RNeasy Plus Universal Mini Kit (catalogue no. 73404; Qiagen). Libraries were made using the Illumina TruSeq Stranded mRNA Library Kit according to the manufacturer's instructions. Libraries were quantified (kappa qPCR), and equimolar pools were sequenced, using the Illumina NextSeq500 (1 × 75 bp, 30 Gb, 400 M SE/800 M PE reads). RNA-Seq, 75 bp single end, libraries were generated from matched patient samples from endometrial scratch ($n = 7$) and menstrual flow treatment ($n = 7$). The raw sequencing data are deposited at EBI-EMBL ArrayExpress with experimental code E-MTAB-9284.

For each library, original reads files were aligned to GRCh38 human genome (Ensembl Release 84) with STAR (v2.5.1b_modified)[29]. Alignments and quality control (QC) were processed using a ClusterFlow[30] (v0.5dev, https://github.com/ewels/clusterflow) with the 'fastqc_star' pipeline. QC reports were assessed using MultiQC[31] (v0.9dev), which includes output from FastQC[32] (v0.11.5), Trim_galore[33] (v0.6.4) and fastq_screen[34] (version 0.9.3). Gene quantification was determined with HTSeq-Counts[35] (v0.6.1p1). Differential gene expression analysis was performed with DESeq2 (v1.26.0) package in R[36] (v3.6.2). Significant differentially expressed genes between Menstrual and Endometrial scratch samples were selected using the threshold for Benjamini–Hochberg (BH) adjusted $p$-value < 0.05 and log2Fold-Change greater than 1.

Initial quality control prior to calculating differential expression was by principal component analysis (PCA). The PCA plot was generated by using the log2 transformation on the raw count data from DESeq2 with the rlogTransformation function with option 'blind=F'. Then the top 2000 most variables genes were selected to perform the PCA analysis (Fig. 2a). Based on the PCA plot, the endometrial scratch and the menstrual blood samples did not form distinct clusters; therefore, a small number of differential expressed genes (DEGs) were identified. Only 101 genes passed the threshold settings with $p_{adj} < 0.05$ and absolute log2Fold-Change greater than 1 (equivalent to a 2-fold-change). Figure 2b shows these 101 DEGs with the log2 transformed counts for each patient sample as a heatmap, with sample clustering. For each individual patient the endometrial scratch and menstrual blood are clustered together. Seven sets of marker genes for menstrual flow organoids are visualised as a heatmap in Fig. 2c using the same log2 transformed counts.

Further details, scripts and expression-count data are available online (https://github.com/CTR-BFX/Cindrova-Davies_Turco).

**Quantitative real-time RT-PCR analysis**. Total RNA was isolated from snap-frozen organoids using RNAeasy kit (Qiagen, Crawley, UK). RNA was quantified by spectrophotometry (Nanodrop Technologies, DE, USA) and integrity assessed using an Agilent 2100 bioanalyser (Agilent Technologies UK Limited, UK). In brief, 20 µg of total RNA from each sample was reverse transcribed using a master mix containing SuperScript II Reverse Transcriptase in the First Strand Buffer with 0.1 M DTT (Invitrogen, Paisley, UK), 50 ng/ml random hexamers (Sigma). The DNA Engine Opticon 2 Sequence Detection System (Bio-Rad Laboratories, UK) was used to perform real-time PCR according to the manufacturer's protocols (using TaqMan–FAM1 dye). Ct values for each transcript were compared with those for 18S rRNA (dCt obtained), and these values were compared to term control samples (ddCt values are reported). TaqMan gene expression assays (Applied Biosystems, ABI, Warrington, UK): PAEP (Hs01046125_m1), LIF (Hs01055668_m1), MUC1 (Hs00159357_m1), 18S (Hs03003631_g1).

Expression levels were calculated applying the comparative Cycle threshold (Ct) method. Relative expression levels were normalised to the geometric mean of three housekeeping genes HPRT1 (Hs02800695_m1), TOP1 (Hs002432257_m1) and TBP (Hs00427620_m1) using Microsoft Office Excel. All qRT-PCR experiments were carried out with a non-template control.

**Western blotting**. Organoids were homogenised in ice-cold lysis buffer (1 ml of buffer per 100 mg tissue) containing 20 mM Tris, pH 7.4, 1 mM EGTA, 0.01 % Triton x100, 1 mM sodium pyrophosphate, 1 mM sodium orthovanadate, 10 mM β-glycerol phosphate, 50 mM sodium fluoride and a complete mini protease inhibitor cocktail (Roche, Roche Diagnostics, East Sussex, UK). Organoid homogenates were centrifuged at $15,000 \times g$, 4 °C for 20 min. Protein concentrations were determined on the supernatant using a BCA protein assay kit (Sigma, Poole, UK). Lysates were mixed with 3x SDS PAGE sample buffer, boiled for 5 min and allowed to return to room temperature. Equal amounts of protein (30 µg) were separated by sodium dodecyl sulphate-polyacrylamide gel electrophoresis, using 7.5–12.5% polyacrylamide resolving gels, and transferred onto nitrocellulose membrane (Invitrogen, Paisley, UK), and subjected to immunoblot analysis. Membranes were blocked for 1 h at 25 °C in 5% milk diluted in Tris-buffered saline (TBS) and 0.1% Tween 20 and incubated with anti-glycodelin (Abcam, ab53289, 1:1000), anti-MUC1 (Abcam, ab28081, 1:500), anti-LIF (Santa Cruz, sc-1336, 1:1000), anti-progesterone receptor (Abcam, ab32085, 1:1000) or β-actin (Invitrogen, 15G5A11/E2) primary antibodies overnight at 4 °C. After washing and incubating with secondary antibodies, immunoreactive proteins were visualised by the ECL plus chemiluminescence system following the manufacturer's instructions

(Amersham Biosciences, Bucks., UK). Protein bands were quantified using Image J software (National Institutes of Health, http://rsb.info.nih.gov/ij/). Protein loading was normalised against β-actin staining. The values are expressed as a percentage of the control lysate (100%) for each experiment. All uncropped scans are presented in Supplementary Figs. 6 and 7.

**Glycodelin ELISA assay**. At the end of each hormonal treatment experiment, culture medium was pooled from four wells per condition and centrifuged at 12,000 RPM for 5 min. Supernatants were transferred to a fresh tube and stored at −80 °C until analysis. Human PP14/glycodelin A ELISA (RayBiotech Inc., ELH-PP14) was performed using 100 μl supernatant in duplicate alongside a series of diluted human glycodelin standard according to the manufacturer's instructions. The concentration of glycodelin in the supernatants was calculated from the line formula of the standard plots.

**Immunohistochemistry**. Organoids were removed from Matrigel using Cell Recovery Solution, fixed in 4% paraformaldehyde and encapsulated into 1% agarose (Melford, MB1200), dehydrated and embedded in paraffin wax. 7 μm thick sections were dewaxed, rehydrated, incubated in 3% $H_2O_2$ for 15 min to block endogenous peroxidase activity. If required, heat-induced antigen retrieval was performed by boiling sections in Tris-EDTA buffer (10 mM Tris Base, 1 mM EDTA Solution, 0.05% Tween 20, pH 9.0) in a microwavable pressure cooker for 1–2 min (microwave at 570 W). Sections were blocked for 1 h in 5% goat serum, 2% BSA solution (in TBS) to prevent non-specific antibody binding and incubated overnight at 4 °C with the appropriate primary antibodies: prolactin receptor (Preprotech, Cat no: 100-07, no retrieval, 1:500), progesterone receptor (Dako, M3569, TE, 1:50), acetylated tubulin (Cell Signaling, #611B1, TE, 1:1000), glycodelin (Abcam, ab53289, TE, 1:500), Ki67 (Abnova, PAB12127, TE, 1:10,000). This was followed by detection with avidin-conjugated secondary antibodies, and visualised with a Vectastain Elite ABC kit (Vector Laboratories, Peterborough, UK) and SigmaFast DAB (Sigma, Poole, UK). Sections were then lightly counterstained with haematoxylin, before being dehydrated and mounted.

**Quantification of the Ki67 proliferation index**. Ki76-stained slides were scanned using a NanoZoomer (Hamamatsu, Welwyn Garden City, UK). Each slide was visualised and four random fields of view were captured per slide at x40 magnification. In each image, the number of Ki67-positive nuclei and the total number of nuclei were counted.

**Haematoxylin and eosin (H&E) staining**. Seven micrometre thick sections were dewaxed, rehydrated and stained in Mayer's haematoxylin for 10 min. They were then washed in running tap water for 15 min and stained in eosin for 4 min, followed by dehydration and mounting.

**Periodic acid Schiff (PAS) staining**. Seven micrometre thick sections were dewaxed, rehydrated and treated with 1% periodic acid (Sigma, 395132) for 15 min, rinsed in water, immersed in Schiff's reagent (Sigma, 3952016) for 15 min, rinsed in water, counterstained with Mayer's haematoxylin for 2 min, washed in running tap water, dehydrated and mounted.

**Processing of organoids for semi-thin sections and lectin staining**. At the end of each experiment, organoids (six wells per treatment) were removed from Matrigel using Cell Recovery Solution, lightly fixed in 1% glutaraldehyde/1% formaldehyde for 1 h, washed with human serum, fixed in 2% glutaraldehyde/2% formaldehyde for 2 h, and washed in cacodylate buffer. The cell pellets were processed into epoxy resin, as previously described[37]. Resin-embedded material was sectioned at 0.5 μm and stained with 1% toluidine blue in 1% borax on a hotplate in order to identify suitable areas for lectin staining. Sections were then cut at 0.75 μm, mounted on APES coated slides and dried at 50 °C for 2 days before staining with *Bandeiraea simplicifolia* -II (BSA-II) and *Arachis hypogaea* agglutinin (AHA) lectin[38]. BSA-II lectin binds to amylase-sensitive glycogen, and sections were treated with 1% amylase for 20 min before staining with BSA-II to detect glycogen[39].

**Statistical analysis**. Data are expressed as mean ± SD. Comparisons were made using a two-tailed Student's *t*-test or ANOVA with a Tukey's multiple comparison post hoc test where appropriate. Differences were considered to be significant at $p \leq 0.05$.

**Reporting summary**. Further information on research design is available in the Nature Research Reporting Summary linked to this article.

## Data availability

All source data underlying the graphs and tables are available in Supplementary Data files and/or can be found in GitHub with corresponding https://doi.org/10.5281/zenodo.4696302 (https://zenodo.org/record/4696302#.YHl2RGjTWjg)[40]. Microcopy images data are available in Cambridge research repository Apollo with DOI link (https://doi.org/10.17863/CAM.68319). All other data (if any) are available upon reasonable request.

## Code availability

The customised scripts used for the analysis can be found in the following online repository: https://github.com/CTR-BFX/Cindrova-Davies_Turco (https://doi.org/10.5281/zenodo.4696302, https://zenodo.org/record/4696302#.YHl2RGjTWjg)[40]. The bioinformatics software used are R version 3.6.2 (corresponding package versions are listed in GitHub).

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

## Acknowledgements

M.Y.T. is supported by the L'Oreal-UNESCO UK and Ireland Fellowship For Women In Science, the Royal Society Dorothy Hodgkin Fellowship (DH160216) and has received funding from the European Research Council (ERC) under the European Union's Horizon 2020 research and innovation programme (Grant agreement No. [853546]). We also thank the Centre for Trophoblast Research for funding this study.

## Author contributions

T.C.D., G.J.B., A.M. and M.Y.T. conceived the idea of isolating endometrial organoids from menstrual flow, and they contributed to writing the manuscript. T.C.D. derived menstrual organoids, performed experiments and wrote the first draft. X.Z. performed RNA-Seq analysis, data processing and figure and manuscript preparation. K.E. facilitated patient consent, tissue and menstrual flow collection. C.J.P.J. performed ultra-structural tissue processing and performed lectin stains.

## Competing interests

The authors declare no competing interests.
