## [Peer Review File · Communications Biology]

Reviewers' comments:

Reviewer #1 (Remarks to the Author):

The paper entitled " The Menstrual flow as a non-invasive source of endometrial organoids ", by Cindrova-Davies and collaborators, is aimed at assessing the effectiveness of menstrual blood glandular organoids in depicting the gene expression profile as well as the response to early pregnancy hormone in comparison with organoids obtained by the endometrial scratch. They reported that the use of menstrual flow as an easily obtainable non-invasive alternative to endometrial biopsies for the derivation of endometrial organoids; indeed, the organoids from the menstrual flow expressed several markers in an identical pattern to organoids derived from the scratch biopsies. As stated by the authors, "Organoids obtained from the two sources were indistinguishable morphologically. In most cases, the initial yield of tissue fragments from MF was lower than that obtained from scratch biopsies. However, the speed of organoid establishment was comparable to that from endometrial biopsies".

Please indicate how do you estimate the number of organoids to evaluate the initial yield and the final recovering.

Authors describe that organoids are stimulated by hormones in order to induce the response of "early pregnancy" This means that the WOI has been fully mimicked and the embryo is able to implant. However, they only use some molecular markers without any morphological assessment. Authors are asked to better define the status of the organoids after hormonal treatment.

Authors also indicate that GdA expression showed a significant increase, as well as the mRNA of LIF and MUC1. But they did not evaluated these at protein levels, whereas at protein level they evaluated other markers. Please The Authors may explain this change between selected genes and proteins.

Reviewer #2 (Remarks to the Author):

This is a succinct manuscript by Cindrova-Davies et al, describing the use of menstrual material as a source to develop endometrial organoids. Applications of this non-invasive method could range from cancer to infections to reproductive disorders. As a non-invasive approach, there is definitely a lot of potential.

General comments:

1) This is a very short paper. I suggest including some of the methodologies into the main text, as establishment of organoids from menstrual flow is a novel procedure that deserves to be adequately explained. More details should also be included for some of the analyses proposed.

2) To improve readability, I suggest the authors start the manuscript from the third paragraph ("Endometrial organoids thus provide etc etc") rather than general human endometrial physiology – as is, the first part of the introduction feels very disjointed from the remainder of the paper.

Specific comments:

a) "We succeeded in deriving organoids from endometrial scratches and menstrual flow (MF) in seven volunteers undergoing an IVF treatment cycle."

What is the take rate? It is helpful to understand whether this is 7/7 or 7/30 patients. Please move up the description of the patients characteristics: is there any difference observed in take rate, morphology or else between the different patients?

b) "Organoids obtained from the two sources were indistinguishable morphologically (Fig. 1b)."

Can the authors show an H&E stain of sections of both, possibly from more than one patient, to substantiate this?

c) "RNA sequencing (RNA-Seq) revealed that organoids from the two sources clustered together (Fig. 1d)."

At what point was the RNASeq performed? Before or after passaging?

d) "RNA sequencing (RNA-Seq) revealed that organoids from the two sources clustered together"

PCA analysis should be explained, both in text and methods. Is it performed on all variable genes or a subset?

e) "There was no evidence of reduced proliferation of MF organoids compared to those derived from biopsies."

Can this be quantified and shown?

f) "In addition, there was a marked increase in protein levels of glycodeilin-A (GdA/PAEP), acetylated tubulin, P4-R, and PRL-R (Fig. 2b-c)."

Firstly, there is very little data provided on the experiment itself. How long was the treatment? Was it performed on passaged organoids? How many replicates / repetitions? Please include statistics for the qRT-PCR data. Can the WB data be quantified?

In addition, there is no context provided to interpret the data, neither here nor in the "discussion" portion of the main text. What is the relevance of the increase in glycodeilin-A (GdA/PAEP), acetylated tubulin, P4-R, and PRL-R?

g) "Menstrual flow can also be used as a source of stromal cells, as we succeeded in deriving stromal cell cultures from the samples."

Unless data is added to support this claim, remove from the manuscript – as is, it is equivalent to "data not shown".

h) "Investigating the physiology and function of the normal endometrium is challenging due to its great variability between individuals, reflecting the profound effects of diverse factors including BMI, age and parity."

Given this, one would expect some differences to be found when comparing organoids of different patients with diverse characteristics. Is this the case?

i) "Equally, co-culture of organoids with immune and stromal cells will allow investigation of the cell-cell interactions important for tissue homeostasis and function, and how these may be altered in disease^{10,11}."

There is no data to support feasibility of co-culturing using menstrual flow-derived samples.

We would like to thank the reviewers and the editor for their helpful and constructive comments. The paper was initially submitted as a brief communication, which restricted us in terms of details. We apologise for the lack of clarity. In addition, we submitted the paper during the lockdown, and, consequently, some of our results lacked in detail, as pointed out by the reviewers. In order to address issues raised by the editor and reviewers, we expanded the methods description, added additional results to characterise the organoids following hormonal treatment, and we rearranged the presentation of the text to make the manuscript more accessible. We also spelt out menstrual flow throughout the text, as recommended. As a consequence of presenting additional data, we now include another author who contributed, Dr Carolyn JP Jones, and we also rearranged the order of the authorship slightly, with Dr Margherita Y Turco now being the senior author. Changes in the manuscript are highlighted in yellow. Please find below our detailed response to individual comments (in blue):

Reviewer #1 (Remarks to the Author):

The paper entitled " The Menstrual flow as a non-invasive source of endometrial organoids ", by Cindrova-Davies and collaborators, is aimed at assessing the effectiveness of menstrual blood glandular organoids in depicting the gene expression profile as well as the response to early pregnancy hormone in comparison with organoids obtained by the endometrial scratch. They reported that the use of menstrual flow as an easily obtainable non-invasive alternative to endometrial biopsies for the derivation of endometrial organoids; indeed, the organoids from the menstrual flow expressed several markers in an identical pattern to organoids derived from the scratch biopsies. As stated by the authors, "Organoids obtained from the two sources were indistinguishable morphologically. In most cases, the initial yield of tissue fragments from MF was lower than that obtained from scratch biopsies. However, the speed of organoid establishment was comparable to that from endometrial biopsies". Please indicate how do you estimate the number of organoids to evaluate the initial yield and the final recovering.

Our response: We performed additional experiments and assays in order to assess organoid formation quantitatively. We now show a detailed time line of menstrual organoid formation (Fig. 1b), we present the single cell assay in which menstrual vs. scratch organoids from 4 patients were seeded from single cells and resulting efficiency of organoid formation quantified (Fig. 1d-e). We also quantified the proliferation rate of paired menstrual vs. scratch organoids from 3 patients, using Ki67 immunostaining (Fig. 1f-g). Methods were expanded to describe these techniques.

Authors describe that organoids are stimulated by hormones in order to induce the response of "early pregnancy" This means that the WOI has been fully mimicked and the embryo is able to implant. However, they only use some molecular markers without any morphological assessment. Authors are asked to better define the status of the organoids after hormonal treatment.

Our response: As advised, we performed additional experiments to define the differentiation status of the organoids after hormonal treatment. These include: quantification of the qPCR results of differentiation markers (Fig. 2d); running and quantification of western blots to detect glycodeilin, MUC-1 and LIF in 5 menstrual vs. 5 scratch organoid samples (Fig. 2e-f); and quantification of glycodeilin secretion in culture supernatants from hormonally-challenged menstrual vs. scratch organoids (n=4 each)(Fig. 2g). We also include more data on P4-R expression (Fig. 3b-d) to explore the

concept of P4-R downregulation in response to progesterone. In addition, we provide more evidence of morphological changes in response to hormonal treatment, using H&E staining (Fig. 3e and Suppl. Fig.3), and lectin staining of resin sections of menstrual vs. scratch-treated organoids (Fig.3f, Suppl. Fig. 1). Overall, these results indicate that the organoids acquire a late secretory phase/early pregnancy phenotype.

Authors also indicate that GdA expression showed a significant increase, as well as the mRNA of LIF and MUC1. But they did not evaluate these at protein levels, whereas at protein level they evaluated other markers. Please The Authors may explain this change between selected genes and proteins.

Our response: We apologise for the inconsistency. As described above, we have now quantified both protein and mRNA expression of MUC-1, glycodelin and LIF. We also measured glycodelin secretion in the culture supernatants.

Reviewer #2 (Remarks to the Author):

This is a succinct manuscript by Cindrova-Davies et al, describing the use of menstrual material as a source to develop endometrial organoids. Applications of this non-invasive method could range from cancer to infections to reproductive disorders. As a non-invasive approach, there is definitely a lot of potential.

General comments:

1) This is a very short paper. I suggest including some of the methodologies into the main text, as establishment of organoids from menstrual flow is a novel procedure that deserves to be adequately explained. More details should also be included for some of the analyses proposed.

Our response: Thanks for this suggestion. We have expanded the methods section and it is now part of the main text.

2) To improve readability, I suggest the authors start the manuscript from the third paragraph ("Endometrial organoids thus provide etc etc") rather than general human endometrial physiology – as is, the first part of the introduction feels very disjointed from the remainder of the paper.

Our response: The paper now starts by describing the organoid system as suggested.

Specific comments:

a) "We succeeded in deriving organoids from endometrial scratches and menstrual flow (MF) in seven volunteers undergoing an IVF treatment cycle."

What is the take rate? It is helpful to understand whether this is 7/7 or 7/30 patients. Please move up the description of the patients characteristics: is there any difference observed in take rate, morphology or else between the different patients?

Our response: We apologise for the lack of detail, as explained above, the initial submission was a brief communication, which was highly restrictive in terms of the word limit. In fact, we performed an initial pilot study (ethical approval HBREC.2017.10) to

optimise the technique of menstrual organoid derivation. We received samples from 7 volunteers and succeeded in deriving menstrual organoids from 6 of these. We had a repeat sample from the unsuccessful volunteer and this also failed, cells looked dead. We accounted this to sample collection/volunteer problem, but we did not explore the issue further. In addition, 3 volunteers in this pilot study provided repeat samples, and we successfully derived organoids from all these samples as well. In order to characterise the menstrual flow organoids, we derived organoids from endometrial scratches and menstrual flow in 7 volunteers undergoing an IVF treatment cycle at the Bourn Hall Clinic. We received menstrual blood samples from 8 patients and succeeded in deriving menstrual flow organoids from 7 of these. In the failed sample, the tissue looked white and dead, organoids did not grow at all. We assumed there was a technical problem with the collection, but this was not explored further. Altogether, the success rate was 87%.

The rate of growth varied between patients, as it does in organoids derived from scratch biopsies³ (Fig. 1b). We were able to freeze menstrual gland digest samples and recover them later (Fig. 1b). Organoids obtained from the two sources were indistinguishable morphologically (Fig. 1c). In most cases, the initial yield of tissue fragments from menstrual flow was lower than that obtained from scratch biopsies. However, the speed of organoid establishment was comparable to that from endometrial biopsies³. In order to compare the growth rate of menstrual vs. scratch organoids, we quantified organoid efficiency formation in paired single cell assays (Fig. 1d-e) and we assessed organoid proliferation rate using Ki67 staining (Fig. 1f-g). There was no difference between paired organoids from menstrual flow vs. scratch biopsy in terms of organoid efficiency formation from single cells (mean \pm SE organoids efficiency formation: 62.67 \pm 19.70 menstrual organoids vs. 57.97 \pm 16.64 scratch organoids formed after 9d from 5,000 single cells seeded per well), or by counting the Ki67 proliferation index (mean \pm SE Ki67 proliferation rate: 27.76 \pm 4.32 % menstrual vs. 28.12 \pm 4.16 % scratch). Menstrual flow organoids began to form after 2-4 days in culture if grown from fresh samples, or after 5 days if grown from frozen digests (Fig. 1b), and were ready to passage after 7-10 days (Fig. 1b).

These important details are now in the Results and Discussion section, and the Methods section has been expanded.

b) "Organoids obtained from the two sources were indistinguishable morphologically (Fig. 1b)."

Can the authors show an H&E stain of sections of both, possibly from more than one patient, to substantiate this?????

Our response: This is a good suggestion, thank you. We provide an H&E staining of hormonally-treated menstrual vs. scratch organoids from N=3 patients (Fig. 3e, Suppl. Fig. 3). In addition, we present lectin staining of resin sections of menstrual vs. scratch-treated organoids, which also clearly illustrates the morphology changes that hormones induce (Fig.3f, Suppl. Fig. 1).

c) "RNA sequencing (RNA-Seq) revealed that organoids from the two sources clustered together (Fig. 1d)."

At what point was the RNASeq performed? Before or after passaging?

Our response: Organoids were harvested using the Cell Recovery Solution without manual disruption, so when the organoids have grown to full size which is around 7-10d

after passaging. RNA was then extracted and RNASeq performed. This is now described in the methods section.

d) "RNA sequencing (RNA-Seq) revealed that organoids from the two sources clustered together"

PCA analysis should be explained, both in text and methods. Is it performed on all variable genes or a subset?

Our response: Initial quality control prior to calculating differential expression is a principal component analysis (PCA). The PCA plot was generated by using the log₂ transformation on the raw count data from DESeq2 with the rlogTransformation function with option "blind=F". Then the top 2000 most variable genes were selected to perform the PCA analysis (Fig 2a). Based on the PCA plot, the endometrial scratch and the menstrual blood samples do not form distinct clusters, therefore a small number of differential expressed genes (DEGs) were identified. Only 101 genes passed the threshold settings with padj < 0.05 and absolute log₂FoldChange greater than 1 (equivalent to a 2-fold change). Fig 2b shows these 101 DEGs with the log₂ transformed counts for each patient sample as a heatmap, with sample clustering. For each individual patient the results from endometrial scratch and menstrual blood are clustered together. Seven sets of marker genes for menstrual flow organoids are visualised as a heatmap in Fig2c using the same log₂ transformed counts.

This has been added to the text and methods.

e) "There was no evidence of reduced proliferation of MF organoids compared to those derived from biopsies."

Can this be quantified and shown? –

Our response: We performed additional experiments and assays in order to assess organoid formation quantitatively. We now show a detailed time line of menstrual organoid formation (Fig. 1b), we present the single cell assay in which menstrual vs. scratch organoids from 4 patients were seeded from single cells and resulting efficiency of organoid formation quantified (Fig. 1d-e), and we also quantified the proliferation rate of paired menstrual vs. scratch organoids from 3 patients, using Ki67 immunostaining (Fig. 1f-g). Methods were expanded to describe these techniques.

f) "In addition, there was a marked increase in protein levels of glycodeilin-A (GdA/PAEP), acetylated tubulin, P4-R, and PRL-R (Fig. 2b-c)."

Firstly, there is very little data provided on the experiment itself. How long was the treatment? Was it performed on passaged organoids? How many replicates / repetitions? Please include statistics for the qRT-PCR data. Can the WB data be quantified?

In addition, there is no context provided to interpret the data, neither here nor in the "discussion" portion of the main text. What is the relevance of the increase in glycodeilin-A (GdA/PAEP), acetylated tubulin, P4-R, and PRL-R?

Our response: As advised, we performed additional experiments. These include quantification of the qPCR results (Fig. 2d), running and quantification of western blots to detect glycodeilin, MUC-1 and LIF in 5 menstrual vs. 5 scratch organoid samples (Fig. 2e-f), quantification of glycodeilin secretion in culture supernatants from hormonally-challenged menstrual vs. scratch organoids (n=4 each)(Fig. 2g). We also include more

data on the P4-R expression (Fig. 3b-d) to explore the concept of P4-R downregulation in response to progesterone. We explore the changes in acetylated tubulin, glycodefin secretion and P4-R and PRL-R expression in more detail in the text.

g) "Menstrual flow can also be used as a source of stromal cells, as we succeeded in deriving stromal cell cultures from the samples." Cross out!

Unless data is added to support this claim, remove from the manuscript – as is, it is equivalent to "data not shown".

Our response: We agree. Whilst we succeed in deriving stromal cells as well, we do not have robust data to show this. We thus crossed out the sentence.

h) "Investigating the physiology and function of the normal endometrium is challenging due to its great variability between individuals, reflecting the profound effects of diverse factors including BMI, age and parity."

Given this, one would expect some differences to be found when comparing organoids of different patients with diverse characteristics. Is this the case?

Our response: Yes, this is indeed the case. We are currently exploring this issue in another study. There are certainly differences, and this can be seen even in the results we present here, e.g. in the single cell assay and Ki67 quantification.

i) "Equally, co-culture of organoids with immune and stromal cells will allow investigation of the cell-cell interactions important for tissue homeostasis and function, and how these may be altered in disease^{10,11}."

There is no data to support feasibility of co-culturing using menstrual flow-derived samples.

Our response: We agree, this statement has been crossed out.

Reviewers' comments:

Reviewer #1 (Remarks to the Author):

The manuscript by Tereza Cindrova-Davies et al. entitled "Menstrual flow as a non-invasive source of endometrial organoids" explores an alternative method to the endometrial biopsy for the retrieval of endometrial organoids. Based on these data, the present study is on a topic of high relevance and general interest to the readers of the journal. The manuscript after the first revision has improved and shows a valid point. Unfortunately, has still many inaccuracies that prevent me to recommend this paper for publication as it stands.

Specific comments:

Line 44: I would recommend expanding your references; there are few groups working on endometrial organoids worth of mention, for example:

-Fitzgerald HC, Dhakal P, Behura SK, Schust DJ, Spencer TE. Self-renewing endometrial epithelial organoids of the human uterus. *Proc Natl Acad Sci U S A*. 2019 Nov 12;116(46):23132-23142. doi: 10.1073/pnas.1915389116. Epub 2019 Oct 30. PMID: 31666317; PMCID: PMC6859318.

-Luddi A, Pavone V, Semplici B, Governini L, Criscuoli M, Paccagnini E, Gentile M, Morgante G, Leo V, Belmonte G, Zarovni N, Piomboni P. Organoids of Human Endometrium: A Powerful In Vitro Model for the Endometrium-Embryo Cross-Talk at the Implantation Site. *Cells*. 2020 Apr 30;9(5):1121. doi: 10.3390/cells9051121. PMID: 32366044; PMCID: PMC7291023.

Line 221-222 The authors describe that the organoids are cryopreserved with five-star medium (please add a reference). Anyway, in the main text, the authors state that menstrual gland "digest" samples were frozen (line 68). The authors have to explain if the experiments are carried out with frozen gland digests or with frozen organoids. In the first case, why the authors state that organoids are cryopreserved with a five-star medium? I guess 3D organoids it's quite delicate culture, did the authors ever thaw them to check the viability? Accordingly, are the organoids showed in Fig 1b frozen/thawed organoids or organoids obtained from frozen/thawed digest glands? The images showed in Figure 1b, c and d are not comparable in terms of magnification. For example, it is not possible that organoids showed in 1b are at the same magnification that 1d! Moreover, we suggest selecting in Fig1b few time points and to show higher magnification, in order to clearly demonstrate the presence of organoids in each sample, since at present magnification only black dots are visible (they look like necrotic bodies).

Line 314: If the authors would indicate they use 30-50 µg of protein for each gel, they have to indicate when 30 or 50µg were loaded.

The normalization procedure used for western blotting analysis (Ponceau-S, see figures) is quite questionable (the authors have to add the corresponding reference). Usually, Actin or GAPDH is used to normalize protein abundance. Moreover, the authors stated that "Protein loading was normalized against β324 actin staining" (line 324). Despite this sentence, none western blots show the actin staining. Therefore, in the revised manuscript the authors must show the actin staining corresponding to each western blot (please, also include the weight protein sizes) Finally, the uncropped WB images must be shown to the editor (maybe as supplementary material).

Reviewer #2 (Remarks to the Author):

The authors have clarified all points raised in the previous version of this work. I believe the added details and analyses strengthen this work further. It is a very interesting study and I commend the authors on their responsiveness to the critiques.

We would like to thank reviewer #1 and the editor for additional helpful comments. We addressed all queries raised and hope this will be satisfactory.

Regarding the summary comments made by the editor:

'In particular, please ensure that the Methods section is sufficiently detailed that it can be reproduced and address the other questions about the methods raised by the reviewer.'

Our answer: We addressed all comments raised by the reviewer (please see below for details) and expanded the methods section.

'Please provide the uncropped, unedited western blot images as a Supplementary Figure, together with some additional representative blot images.'

Our answer: All uncropped blots used for the quantification are now presented in Suppl. Figs. 6 and 7.

'Please also ensure that all blot images in the main figures include loading controls and size markers.'

Our answer: We now provide β -actin as a loading control, and we show size markers on all uncropped scans of western blots.

'Finally, I strongly urge you to consider separating the Results and Discussion sections. In either case, please add topical subheadings to the main Results section (1 subheading per figure is a general guideline).'

Our answer: Thanks for this suggestion. We separated the Results and Discussion sections, and added topical subheadings in the main Results sections.

'Please highlight all changes in the manuscript text file.'

Our answer: All changes are highlighted in the manuscript text file.

Regarding the summary comments made by Reviewer #1:

The manuscript by Tereza Cindrova-Davies et al. entitled "Menstrual flow as a non-invasive source of endometrial organoids" explores an alternative method to the endometrial biopsy for the retrieval of endometrial organoids. Based on these data, the present study is on a topic of high relevance and general interest to the readers of the journal. The manuscript after the first revision has improved and shows a valid point. Unfortunately, has still many inaccuracies that prevent me to recommend this paper for publication as it stands.

Specific comments:

Line 44: I would recommend expanding your references; there are few groups working on endometrial organoids worth of mention, for example:

-Fitzgerald HC, Dhakal P, Behura SK, Schust DJ, Spencer TE. Self-renewing endometrial epithelial organoids of the human uterus. Proc Natl Acad Sci U S A. 2019 Nov 12;116(46):23132-23142. doi: 10.1073/pnas.1915389116. Epub 2019 Oct 30. PMID: 31666317; PMCID: PMC6859318.

-Luddi A, Pavone V, Semplici B, Governini L, Criscuoli M, Paccagnini E, Gentile M, Morgante G, Leo V, Belmonte G, Zarovni N, Piomboni P. Organoids of Human Endometrium: A Powerful In Vitro Model for the Endometrium-Embryo Cross-Talk at the Implantation Site. Cells. 2020 Apr 30;9(5):1121. doi: 10.3390/cells9051121. PMID: 32366044; PMCID: PMC7291023.

Our answer: Thank you for this reminder. We added these references and we also discuss some of them in the text.

Line 221-222 The authors describe that the organoids are cryopreserved with five-star medium (please add a reference). Anyway, in the main text, the authors state that

menstrual gland “digest” samples were frozen (line 68). The authors have to explain if the experiments are carried out with frozen gland digests or with frozen organoids. In the first case, why the authors state that organoids are cryopreserved with a five-star medium? I guess 3D organoids it's quite delicate culture, did the authors ever thaw them to check the viability? Accordingly, are the organoids showed in Fig 1b frozen/thawed organoids or organoids obtained from frozen/thawed digest glands?

Our answer: Our apologies if our method description raised any undue concerns, this was not intended. The 'five-star' medium is a in-house name for a freezing medium tested for primary cell and organoid cultures in our lab. It contains 70% organoid culture medium, 20% FBS and 10% DMSO, as described in the methods. We dropped out the word five-star not to confuse the reader.

Regarding cryopreservation, we cryopreserve and freeze all our organoid cultures, and we biobank them, at all passages, starting from gland/menstrual digests (passage 0) to higher passages. The viability of the frozen cultures reflects that of the cultures before freezing. For all experiments presented in this paper, we thawed established organoid cultures (not frozen digests) of a low passage (1-3), expanded their numbers and used experimentally. The viability of the frozen cultures reflects that of the cultures before freezing. This has been clarified in the methods.

The derivation of organoids from frozen menstrual and gland digests was only presented in Fig. 1b and Suppl. Fig. 1 where we not only document the speed of derivation of organoids from fresh menstrual blood (top panel), but we also present the growth of organoids from frozen menstrual and scratch digests from the same patient (middle and bottom panel). This was to illustrate the versatility of the technique, i.e. one can process organoids from menstrual blood or endometrial scratch and cryopreserve these digests/passage 0 cultures, to be able to grow them and expand at a later date. In this instance, the frozen menstrual and scratch digests (middle and bottom panel) were actually derived from one patient, as we could not provide this evidence from the same patient using fresh samples. The speed of recovering organoids from these two sources in the same patient are remarkably similar. We did label the differences between these samples on the LHS of each panel, and explained this in the original figure legend.

As per the reviewer's recommendation below, we changed Fig. 1b to show images of higher magnification that better reflect the growth of the organoids. The original low magnification images are now in Suppl. Fig. 1.

The images showed in Figure 1b, c and d are not comparable in terms of magnification. For example, it is not possible that organoids showed in 1b are at the same magnification that 1d!

Our response: With all due respect, Fig. 1c shows a 200 μm scale bar, while the original Fig. 1b (now Suppl. Fig. 1) and Fig. 1d show a scale bar of 2,000 μm . We did not make this up. You can see that Suppl. Fig. 1 (formally Fig. 1b) and Fig. 1d are of the same magnification, as they show the whole Matrigel drop per image, which can only be visualized when we use our x2 lens (2,000 μm scale bar). The differences reflect the size of the organoid growth, i.e. organoids in Fig. 1d are of higher passages (both menstrual and scratch organoids of patient 1 and 2 were at passage 2, patient 3 at passage 6 and patient 4 at passage 4). In addition, the organoids were expanded and not derived straight from the digest as we let Fig. 1d organoids expand to a large extent for 9d so that we could then count the formation of organoids from single cells in this assay. This reflects the differences in the size of the actual organoids between the two figures.

Moreover, we suggest selecting in Fig1b few time points and to show higher magnification, in order to clearly demonstrate the presence of organoids in each sample, since at present magnification only black dots are visible (they look like necrotic bodies).

Our response: We agree, and as mentioned above, we changed Fig. 1b to show fewer images of higher magnification that better reflect the growth of the organoids. The original low magnification images are now in Suppl. Fig. 1.

Line 314: If the authors would indicate they use 30-50 μg of protein for each gel, they have to indicate when 30 or 50 μg were loaded.

Our response: We apologise, this information was copied straight from our generic western blot methodology, not reflecting the organoids specifically. Protein assay was performed for each organoid experiment and 30 μg of protein was loaded for each gel. We clarified this in the methods section.

The normalization procedure used for western blotting analysis (Ponceau-S, see figures) is quite questionable (the authors have to add the corresponding reference). Usually, Actin or GAPDH is used to normalize protein abundance. Moreover, the authors stated that "Protein loading was normalized against β 324 actin staining" (line 324). Despite this sentence, none western blots show the actin staining. Therefore, in the revised manuscript the authors must show the actin staining corresponding to each western blot (please, also include the weight protein sizes).

Finally, the uncropped WB images must be shown to the editor (maybe as supplementary material).

Our answer: In our defence, we used Ponceau S as a loading control in the original submission because it has been validated by Romero-Calvo et al. (Analyt. Biochem. 2010; 401: 318–320). These authors applied reversible Ponceau staining to check equal loading of gels and measured actin in parallel under different conditions. They demonstrated that densitometric analysis was comparable with both techniques, and thus validated routine quantitation of Ponceau staining before antibody probing as an alternative to β -actin blotting. However, as suggested by the reviewer, we also run β -actin as a house keeping protein, and we reanalysed the data, using β -actin to normalise for protein loading. These data are now presented in the amended figures.

All uncropped WB images and weights of protein sizes are included in Suppl. Figs. 6-7.